# IMPROVING MEDICAL VISUAL REINFORCEMENT FINE-TUNING VIA PERCEPTION AND REASONING AUGMENTATION

## ABSTRACT

While recent advances in Reinforcement Fine-Tuning (RFT) have shown that rule-based reward schemes can enable effective post-training for large language models, their extension to cross-modal, vision-centric domains remains largely underexplored. This limitation is especially pronounced in the medical imaging domain, where effective performance requires both robust visual perception and structured reasoning. In this work, we address this gap by proposing *VRFT-Aug*, a visual reinforcement fine-tuning framework tailored for the medical domain. VRFT-Aug introduces a series of training strategies designed to augment both perception and reasoning, including prior knowledge injection, perception-driven policy refinement, medically informed reward shaping, and behavioral imitation. Together, these methods aim to stabilize and improve the RFT process. Through extensive experiments across multiple medical datasets, we show that our approaches consistently outperform both standard supervised fine-tuning and RFT baselines. Moreover, we provide empirically grounded insights and practical training heuristics that can be generalized to other medical image tasks. We hope this work contributes actionable guidance and fresh inspiration for the ongoing effort to develop reliable, reasoning-capable models for high-stakes medical applications.

## 1 INTRODUCTION

Recently, Reinforcement Learning (RL)-based fine-tuning Jaech et al. (2024); Shao et al. (2024); Guo et al. (2025); Team et al. (2025); Lightman et al. (2023); Chen et al. (2024a) for large language models (LLMs) has shown significant progress in complex reasoning tasks. The emergence of methods such as DeepSeek-R1 Guo et al. (2025) and the GRPO Shao et al. (2024) algorithm has demonstrated the feasibility of fine-tuning large models using rule-based rewards Lambert et al. (2024); Team et al. (2025) instead of learned reward models Liu et al. (2024); Ouyang et al. (2022); Zang et al. (2025), substantially lowering the barrier to applying RL in large-scale model training and introducing a promising new paradigm. While RL-based fine-tuning has been actively explored in LLMs, its application to large vision-language models (LVLMs) Wang et al. (2024b); Bai et al. (2025a); Chen et al. (2024b)—referred to as Visual Reinforcement Fine-Tuning (V-RFT) Liu et al. (2025); Shen et al. (2025b); Tan et al. (2025); Li et al. (2025)—remains largely underexplored.

Despite its promise, the effectiveness of V-RFT remains constrained by fundamental challenges in visual perception and reasoning. First, a pretrained LVLMs may lack the capacity to capture subtle visual cues or localize key regions without explicit supervision Wang et al. (2024a). This leads to unreliable or sparse rewards during early-stage exploration, hindering stable policy updates Andrychowicz et al. (2018); Devin et al. (2016); Pinto & Gupta (2016). Second, many vision-language tasks require multi-step reasoning Zhao et al. (2024) or structured decision-making, which cannot be effectively learned through scalar reward signals alone. Without explicit reasoning supervision or prior knowledge, V-RFT models are prone to shortcut learning or shallow pattern memorization Amodei et al. (2016), rather than developing genuine reasoning ability. These limitations highlight a pressing need to enhance V-RFT with augmented perception and reasoning mechanisms, enabling more robust learning in visually and cognitively demanding tasks. The gap is even more pronounced in the context of medical imaging domain, where it is still unclear how to effectively perform RL post-training on pretrained LVLMs to improve their clinical utility and generalization.

Before delving into the technical details, we highlight a key distinction between medical image recognition and general-domain vision tasks—an insight that forms the cornerstone of our work. **Specifically, we find that successful medical image understanding hinges on the fusion of perception and reasoning, rather than relying on either in isolation.** The former emphasizes how information is received and interpreted, while the latter focuses on how information is organized, abstracted, and logically manipulated. Perceptual tasks are characterized by their reliance on accurate interpretation of sensory input—once the content of an image is clearly perceived, further analysis may require little to no reasoning. This is exemplified by many Visual Question Answering (VQA) benchmarks Gurari et al. (2018); Goyal et al. (2017); Lin et al. (2015) in the general domain, where models are asked about attributes, positions, or colors of objects. As long as the model can correctly parse the visual elements, it can answer such questions without needing to perform complex inference. In contrast, reasoning tasks Salewski et al. (2022); Zhang et al. (2019) demand an additional layer of logical composition. They require the model to synthesize multiple pieces of information to arrive at a coherent, logically grounded conclusion. Unlike natural images, medical images are not readily interpretable by untrained individuals. Recognizing subtle patterns such as tumors on a CT scan—and further judging their malignancy—often requires both perceptual decoding of the visual content and the integration of domain-specific knowledge Menze et al. (2015). The task thus involves both visual pattern recognition (perception) and medical reasoning based on those patterns.

This naturally gives rise to a central question: *Can reinforcement learning—originally envisioned as a tool to enhance reasoning capabilities—effectively address tasks that require a hybrid of perception and reasoning, such as medical image understanding?* In this work, we take a step toward answering this question by proposing VRFT-Aug, a visual reinforcement fine-tuning framework tailored for the medical domain. VRFT-Aug introduces a series of improvements aimed at two core challenges:

1. Augment LVLM expertise perception capability by dual-channel knowledge injection.

2. Augment LVLM medical reasoning skill by reward shaping.

To achieve these goals, we systematically investigate how RL techniques—specifically GRPO, used as our baseline V-RFT method—can be adapted and extended to better support perception and reasoning in visually and cognitively demanding medical tasks.

**Perception Augmentation via Knowledge Injection.** Because medical image recognition requires extensive domain-specific prior knowledge Jiang et al. (2024); Gao et al. (2024); Wu et al. (2023a); Qin et al. (2022); Yang et al. (2025), we first propose a pipeline that injects such knowledge into pretrained models through both explicit and implicit mechanisms. To address this, we propose a method for explicitly injecting medical knowledge into the model by prompt engineering to enhance its ability to recognize and distinguish domain-specific entities. Inspired by Qin et al. (2022); Yang et al. (2025); Wu et al. (2023b), we introduce the visual attributes–such as color, shape, and location– to the prompts of a medical concept. And the prompt will incentivize the LVLM to recognize objects that share identical visual attributes. Then we propose an implicit method for knowledge injection by exploring cross-task training, leveraging diverse medical vision tasks to encourage transferability and robust generalization. This enables the model to acquire both local (e.g., lesion boundary) and global (e.g., anatomical structure) understanding, crucial for handling the multi-scale nature of clinical reasoning.

**Reasoning Augmentation via Reward Shaping.** Prior studies suggest that the hallucinated content in the reasoning process on the language side leads to incorrect output for tasks like VQA and image captioning Min et al. (2025); Sun et al. (2025); Zhou et al. (2024). This observation suggests that the reasoning process may influence the perceived content during the text decoding process. **Firstly, drawing inspiration from human cognitive mechanisms, we explore whether enforcing repeated recitation of expressive descriptions of medical concepts, as specified in the prompts, could help mitigate hallucinations and guide the model toward more accurate conclusions.** Interestingly, our empirical observations reveal a nuanced outcome. While such repetition during the model's internal reasoning (analogous to a human's internal monologue) can indeed accelerate convergence to a sub-optimal plateau, it often fails to achieve optimal performance in the long run. This suggests that, despite some shared linguistic structures, large models do not always benefit from human-inspired heuristics in the same way, and over-reinforcing certain patterns may limit the model's flexibility and generalization. **Secondly, we design a specialized reward function based on multi-grade fuzzy scheme tailored for ordinal classification tasks commonly found in the**

**medical domain, aiming to help the model distinguish subtle inter-class differences and mitigate the sparse reward problem during early-stage exploration.** By providing more nuanced feedback, the designed reward promotes stable learning and supports the development of accurate reasoning patterns in fine-grained classification tasks.

## 2 RELATED WORKS

**Large Vision Language Models** Large Vision-Language Models (LVLMs) are an evolution of traditional Vision-Language Models (VLMs) Radford et al. (2021); Li et al. (2022b;a), integrating powerful LLMs Achiam et al. (2023); Touvron et al. (2023); Bai et al. (2023; 2025b) with advanced visual perception backbones. This fusion enhances multimodal understanding and complex reasoning across text and visual data, making LVLMs a key step toward Artificial General Intelligence (AGI). LVLMs are categorized into two main types: commercial closed-source models accessible via APIs (e.g., GPT-4o Hurst et al. (2024), Gemini Team et al. (2023; 2024)) and open-source models available for local deployment (e.g., LLaVA Liu et al. (2023), InterVL Chen et al. (2024b), Qwen VL Wang et al. (2024b); Bai et al. (2025a)). The rapid growth of open-source communities has accelerated progress in medical LVLMs, with notable examples like LLaVA-Med Li et al. (2023), developed from LLaVA, and MedRegA Wang et al. (2024a), built on InterVL with continued medical pre-training. Our experiments are based on the advanced Qwen 2.5 VL model.

**Reinforcement Learning** OpenAI's o1 Jaech et al. (2024) pioneered using reinforcement learning (RL) to enhance model reasoning, introducing the test-time scaling law. DeepSeek R1 Guo et al. (2025) extended this with GRPO Shao et al. (2024) and rule-based rewards, becoming the first open-source model to replicate o1's complex reasoning, sparking interest in LLM reasoning research Peng et al. (2025); Muennighoff et al. (2025). In the LVLM domain, R1-V achieved superior performance with GRPO, while VisualThinker-R1-Zero Zhou et al. (2025) showed that applying R1 to base VLMs led to "visual aha moments". MM-Eureka Meng et al. (2025) observed similar effects using RLOO Ahmadian et al. (2024), and Vision-R1 Huang et al. (2025b) introduced a multimodal CoT dataset for enhanced training. Curr-ReFT Deng et al. (2025) proposed a three-stage RL framework. Visual-RFT Liu et al. (2025) uniquely focused on RL for visual perception, while VLM-R1 Shen et al. (2025b) validated R1-style RL across diverse visual tasks. MedVLM-R1 Pan et al. (2025), Med-RLVR Zhang et al. (2025), and Med-R1 Lai et al. (2025) extended RL to the medical domain. Building on these advances, we optimized RL for medical vision with enhanced perception and reward mechanisms.

## 3 METHODS

### 3.1 PRELIMINARY

Visual Reinforcement Fine-Tuning (V-RFT) fine-tunes pretrained LVLMs using reinforcement learning techniques such as PPO Schulman et al. (2017) or GRPO Shao et al. (2024); Guo et al. (2025), enhancing their decision-making capabilities through task-specific, rule-based reward functions (e.g., classification accuracy or IoU). For a downstream task dataset $D$ consisting of $N$ samples, each sample is defined by an input prompt $P$ and its corresponding image $I_i$, where $i$ represents the index of the current sample. The policy model $\pi_\theta$ generates a response $O_i$, which is then evaluated using a rule-based reward function $R$ with respect to the task ground truth $G_i$. Formally, V-RFT aims to optimize the following objective:

$$\max_{\pi_\theta} \frac{1}{N} \sum_{i=1}^{N} \mathbb{E}_{O \sim \pi_\theta(P,I)} R_{\text{V-RFT}}(P, I_i, G_i)$$

$$= \max_{\pi_\theta} \frac{1}{N} \sum_{i=1}^{N} \mathbb{E}_{O \sim \pi_\theta(P,I)} \left[ R(\pi_\theta(O_i \mid P, I_i), G_i) - \beta \, \text{KL} \left[ \pi_\theta(O_i \mid P, I_i) \| \pi_{\text{ref}}(O_i \mid P, I_i) \right] \right], \quad (1)$$

where $\pi_{\text{ref}}$ is the reference model before optimization, and $\beta$ is a hyperparameter controlling the impact of KL-divergence. The rule-based reward function $R$ is defined as:

$$R(\pi_\theta(O_i \mid P, I_i), G_i) = \begin{cases} 1.0 & \text{if } O_i == G_i \text{ or } \text{IoU}(O_i, G_i) \geq threshold, \\ 0.0 & \text{otherwise.} \end{cases} \quad (2)$$

where IoU is the intersection over union metric, and the $threshold$ is typically set to 0.5 as default.

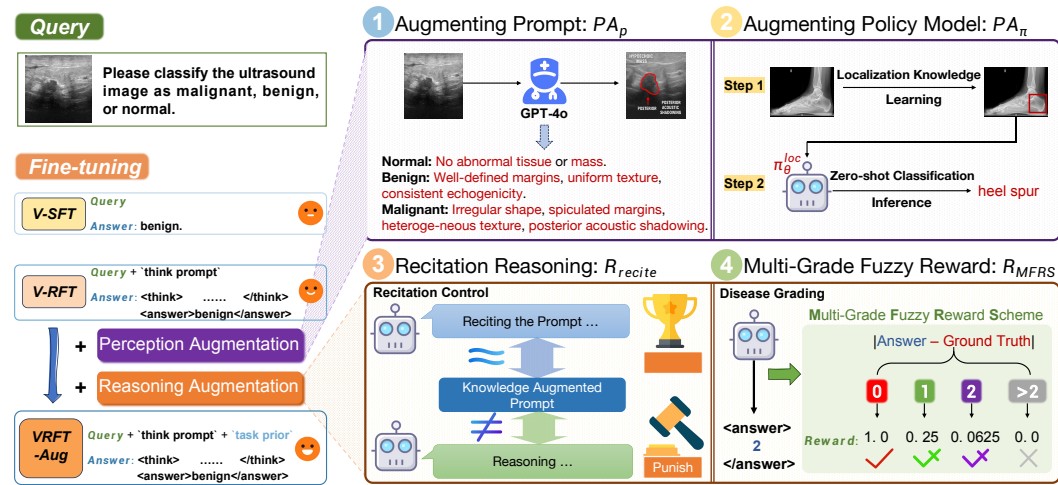

Figure 1: **Overview of VRFT-Aug**. VRFT-Aug incorporates enhancements from both Perception and Reasoning perspectives, introducing four improvement strategies for medical vision tasks: Augmenting Prompt ($PA_p$), Augmenting Policy Model ($PA_\pi$), Recitation Reasoning ($R_{\text{recite}}$), and Multi-Grade Fuzzy Reward ($R_{\text{MFRS}}$).

To improve V-RFT with perception and reasoning capabilities in the medical domain, we propose optimizing Eq. (1) by augmenting its three key components: the prompt $P$, the policy model $\pi_\theta$, and the reward function $R$. For **perception augmentation**, we apply contextual augmentation through a structured prompt $\hat{P}$ (Section 3.2), and implicit knowledge injection by refining the policy $\hat{\pi}_\theta$ (Section 3.3). For **reasoning augmentation**, we adopt reward shaping to guide the learning process: $R_{\text{recite}}$ is designed to capture the model's recitation pattern (Section 3.4), while $R_{\text{MFRS}}$, a task-specific reward based on multi-grade fuzzy reward scheme, is proposed to address the sparse reward problem in medical-grade classification and improve learning effectiveness (Section 3.5).

## 3.2 AUGMENTING PROMPT $P$ WITH TASK-RELEVANT CONTEXT

Pretrained LVLMs often struggle with medical tasks due to the lack of understanding of domain-specific concepts, which are essential for accurate recognition and reasoning. To address this, we first seek to enhance the model's comprehension of medical tasks by expanding the prompt with task-relevant contextual information.

Inspired by prior works in prompt engineering Wang et al. (2023); Denner et al. (2024); Qin et al. (2022), we enrich prompts with visual attributes—such as color, shape, and spatial location—associated with specific medical concepts, therby encouraging the LVLM to focus on relevant objects and strengthening its task-specific perception. To achieve this, we leverage advanced foundation models, such as GPT-4o, to generate relevant visual attributes and create a structured prompt template enriched with task-specific contextual information.

Specifically, for each task, we query GPT-4o with detailed task information—including data source, imaging modality, sample size, and categories—and provide representative images $I_c$ for each category $C$. We then extract comprehensive visual attribute descriptions that capture key aspects essential for solving the task, which we define as explicit contextual knowledge $K_c$. To overcome hallucinations, we manually refine the outputs by consulting medical literature and validating them with medical professionals to ensure clinical accuracy. This contextual knowledge is then used to augment the original input prompt, forming an enhanced prompt $\hat{P}$:

$$\hat{P} = [P, \sum_C K_c] = [P, \sum_C M_{\text{GPT}}(C, I_c)], \tag{3}$$

where $C$ denotes the category of the task to which the image belongs.

Expanding the prompt with task-specific contextual information enhances model performance by providing richer, more relevant cues.This augmented context serves as perceptual guidance, enabling the model to make more accurate predictions. From a theoretical perspective, since the policy

$\pi_\theta(a \mid I, p)$ is conditioned on the prompt $p$, choosing a more informative prompt $p_{\text{rich}}$ results in an initial policy that is closer to the optimal policy $\pi^*$:

$$KL(\pi^* \parallel \pi(\cdot \mid I, p_{\text{rich}})) < KL(\pi^* \parallel \pi(\cdot \mid I, p_{\text{naive}})) \tag{4}$$

This alignment reduces the exploration burden and improves sample efficiency.

### 3.3 AUGMENTING POLICY MODEL $\pi_\theta$ WITH TASK RELEVANT KNOWLEDGE

Beyond enhancing LVLMs with contextual information through prompts ($P$), we further explore whether the policy model ($\pi_\theta$) can transfer knowledge from other relevant tasks through RL, thereby enhancing its perception capability accumulated from cross-task learning.

Inspired by the cognitive workflow of radiologists—"localize first, diagnose later" Litjens et al. (2017); Fan et al. (2024)—we employ the RFT framework to train the model to localize specific regions, lesions, or organs in medical images. These localization priors allow the model to focus its attention on anatomically relevant areas, and thus enhance the perception capability by ruling out irrelevant areas.

Concretely, for medical image classification tasks, we first train the model with a reinforcement learning objective to localize potential regions of abnormality using a small number of samples ($M < N$). During this stage, only a coarse anatomical region is provided as the grounding reference. The model is tasked with predicting a bounding box coordinate $[x_1, y_1, x_2, y_2]$, without receiving any classification-related information. We denote the model that acquired the localization knowledge as $\pi_\theta^{\text{loc}}$ and this implicit knowledge injection process is formulated as:

$$\hat{\pi}_\theta = \pi_\theta^{\text{loc}} = \max_{\pi_\theta} \frac{1}{M} \sum_{i=1}^{M} \mathbb{E}_{O \sim \pi_\theta(P,I)} R_{\text{V-RFT}}(P^{\text{loc}}, I_i, G_i). \tag{5}$$

where $P^{\text{loc}}$ is the prompt designed for localization (more details in the Appendix B.5). Then we use the $\hat{\pi}_\theta$ as the base model to perform zero-shot inference for predicting classification labels $\hat{y}_i^{\text{cls}}$:

$$\hat{y}_i^{\text{cls}} = \hat{\pi}_\theta(O_i^{\text{cls}} \mid P^{\text{cls}}, I_i). \tag{6}$$

### 3.4 AUGMENTING REWARD $R$ WITH RECITATION REASONING

During our experiments on contextual augmentation (Section 3.2), we notice that the model's generated reasoning outputs often appear to recite the medical prior knowledge we implanted in the prompts, a phenomenon we refer to as "Recitation Reasoning". **This observation closely resembles a stereotypical human behavior: when attempting to recognize an unfamiliar concept, humans often reinforce their understanding by mentally or verbally repeating its defining characteristics.** We hypothesize that mimicking this repetitive pattern—by recite medical descriptors throughout the model's internal reasoning steps—can help stabilize attention and output consistency.

To investigate the impact of recitation reasoning, we augment the reward function $R$ with a recitation reward component $R_{\text{recite}}$, enabling us to study this behavior during training by encouraging or discouraging it through reward shaping. Specifically, we hire the Bilingual Evaluation Understudy (BLEU) Papineni et al. (2002) score—a widely adopted metric in natural language generation—to measure the similarity between the model's reasoning outputs and the prior medical knowledge provided in the prompt $\hat{P}$. A higher BLEU score indicates greater repetition of prior knowledge in the output, resulting in a higher recitation reward $R_{\text{recite}}$. The formulation of $R_{\text{recite}}$ is defined as follows:

$$R_{\text{recite}} = \delta \times \text{BLEU}(O_i, \hat{P}), \quad R_{\text{recite}} \in (-1, 1). \tag{7}$$

Following previous work Guo et al. (2025); Huang et al. (2025a); Shen et al. (2025a), we also include accuracy reward $R_{\text{accuracy}}$ and format reward $R_{\text{format}}$. Aggregating these, we obtain the following formula for calculating the overall reward:

$$\hat{R} = \lambda \times R_{\text{accuracy}} + (1 - \lambda) \times R_{\text{format}} + R_{\text{recite}}, \quad \lambda \in (0, 1). \tag{8}$$

where $\lambda$ is a weighting parameter. We control the influence of the recitation reward by adjusting the sign of $\delta$: a positive $\delta$ encourages repetition by rewarding it, while a negative $\delta$ penalizes repetition, which we hypothesize enhances reasoning by stabilizing attention and promoting more independent reasoning.

### 3.5 AUGMENTING REWARD $R$ WITH MULTI-GRADE FUZZY APPROACH

In clinical diagnosis, lesions often differ subtly between adjacent disease grades, with progression marked by gradual changes in quantity, distribution, or extent rather than abrupt shifts. These subtle visual cues make learning difficult and data-intensive. For instance, mild to moderate retinal lesions may differ only slightly in features like microaneurysm count or hemorrhage extent, making them challenging to distinguish Sadda et al. (2020). In early-stage exploration, RL algorithms may suffer from training collapse in particularly challenging tasks where rewards are infrequent—a well-known issue referred to as the sparse reward problem Rengarajan et al. (2022); Dawood et al. (2023). Similarly, when a model fails to detect subtle visual differences, it may make near-correct predictions in grade classification without receiving any reward, further exacerbating learning difficulty and hindering the development of accurate reasoning patterns.

Inspired by multi-objective reward design in reinforcement learning Yang et al. (2024), we introduce a **M**ulti-grade **F**uzzy **R**eward **S**cheme (MFRS) tailored for overcoming the sparse reward problem in medical grading tasks. Specifically, we calculate the difference between the predicted output $O^{\mathrm{cls}}$ and the ground truth $G^{\mathrm{cls}}$, where both $O^{\mathrm{cls}}$ and $G^{\mathrm{cls}}$ are integers labels, and design a "fuzzy" reward mechanism that allows for a relaxed reward even when the predicted value is incorrect. The fuzzy reward weights are selected based on extensive early-stage experiments, as shown in the following formula:

$$R_{\mathrm{MFRS}} = \begin{cases} 1.0 & \text{if } O^{\mathrm{cls}} == G^{\mathrm{cls}}, \\ \frac{1}{4} & \text{if } \mathrm{abs}(O^{\mathrm{cls}} - G^{\mathrm{cls}}) = 1, \\ \frac{1}{16} & \text{if } \mathrm{abs}(O^{\mathrm{cls}} - G^{\mathrm{cls}}) = 2, \\ 0.0 & \text{otherwise.} \end{cases} \tag{9}$$

Therefore, the overall reward is calculated through a weighted average of the updated accuracy reward $R_{\mathrm{MFRS}}$ and the format reward $R_{\mathrm{format}}$. The specific formula is as follows:

$$\hat{R} = \alpha \times R_{\mathrm{MFRS}} + \gamma \times R_{\mathrm{format}}, \tag{10}$$

where $\alpha$ and $\gamma$ are weighting parameters, set to $\alpha = 0.9$ and $\gamma = 0.1$ in this work. As a reward shaping strategy, MFRS works well for medical grade classification tasks and significantly increases the reasoning performance of the model, compared with the Vanilla RFT methods.

The four components are integrated based on task types and training stages: $\mathrm{PA}_P$ (**P**erception **A**ugmentation through **P**rompt) is used in all training pipelines, $\mathrm{PA}_\pi$ is optimized with GRPO for tasks involving object-level alignment, $R_{\mathrm{recite}}$ mitigates over-repetition in reasoning tasks, and $R_{\mathrm{MFRS}}$ is applied for ordinal classification with soft thresholds.

## 4 EXPERIMENTS

### 4.1 SETUP

**Datasets.** To evaluate the effectiveness of our proposed VRFT-Aug in the medical vision domain, we curate datasets from public sources across three representative task types: 1. **Medical Image Classification**, which involves distinguishing anatomical structures or lesions; 2. **Fine-Grained Regional Classification**, targeting the recognition of lesion subtypes within specific anatomical regions; and 3. **Disease Grading**, which assesses both the presence and progression of the disease. We utilize eight datasets from MedMNIST Yang et al. (2023), covering diverse imaging modalities such as X-ray, ultrasound, and CT, to comprehensively evaluate medical image classification tasks. For fine-grained regional classification, we adopt the HAM10000 Tschandl et al. (2018) and Heel Taher & Özacar (2024) datasets. To assess disease progression, we use RetinaMNIST from MedMNIST and the processed COVID-19 dataset Danilov et al. (2022). Detailed information of the datasets can be found in the Appendix B.1.

**Implementation Details.** For both medical image classification and localization tasks, we employ Qwen2.5-VL-3B-Instruct Bai et al. (2025a) as our base reasoning model. Following previous work Shen et al. (2025a); Zheng et al. (2024; 2025); Sheng et al. (2024), we implement the code in Pytorch using 2 NVIDIA A800 80G GPUs. During the RL training, we adopt default GRPO settings, with N set to 8, temperature to 0.9, and KL divergence ratio $\beta$ to 0.04. For the classification task, the model is fully fine-tuned for 120 steps, using a batch size of 256 and the AdamW optimizer with an initial learning rate of 1e-6 for both SFT and RL. For the localization task, the model is fully fine-tuned for

Table 1: Comparison of The best results are highlighted in bold, while the second-best results are underlined. Note that with the exception of RetinaMNIST adopting MFRS reward, all other datasets utilize accuracy reward.

| Shot | Method | Breast | Pneumonia | OCT | Retina* | Derma | Tissue | Blood | OrganA | Average |
|------|--------|--------|-----------|-----|---------|-------|--------|-------|--------|---------|
| 0-shot | Qwen2.5VL-3B | 26.92 | 52.88 | 25.39 | 14.75 | 19.60 | 8.40 | 12.30 | 9.80 | 21.25 |
| | Qwen2.5VL-7B | 8.33 | 39.90 | 33.59 | 21.00 | 21.84 | 10.54 | 7.42 | 8.81 | 18.93 |
| 10-shot | V-SFT | 46.15 | 55.12 | 31.25 | 18.50 | 26.89 | 11.71 | 33.39 | 33.23 | 32.03 |
| | V-RFT | 60.89 | 51.28 | 29.68 | 27.00 | 27.17 | 12.50 | 43.35 | 29.40 | 35.16 |
| | V-RFT + PA$_P$ | **61.53** | **64.42** | **51.17** | **31.75** | **31.92** | **17.18** | **48.04** | **30.39** | **42.05** |
| | Δ | ↑ 0.64 | ↑ 13.14 | ↑ 21.49 | ↑ 4.75 | ↑ 4.75 | ↑ 4.68 | ↑ 4.69 | ↑ 0.99 | ↑ 6.89 |
| 20-shot | V-SFT | 58.33 | 52.24 | 33.59 | 26.75 | 35.01 | 13.08 | 45.31 | 36.22 | 37.57 |
| | V-RFT | 57.69 | 68.42 | 45.31 | 25.00 | 39.49 | 12.50 | 44.33 | 30.82 | 40.45 |
| | V-RFT + PA$_P$ | **67.94** | **72.91** | **55.46** | **37.25** | **40.05** | **17.77** | **54.68** | **38.63** | **48.09** |
| | Δ | ↑ 10.25 | ↑ 4.49 | ↑ 10.15 | ↑ 12.25 | ↑ 0.56 | ↑ 5.27 | ↑ 10.35 | ↑ 7.81 | ↑ 7.64 |
| 256-shot | V-SFT | 40.38 | 71.15 | 44.14 | 50.00 | 45.93 | 19.33 | 59.96 | 37.92 | 46.10 |
| | V-RFT | **73.07** | 81.89 | 70.31 | 59.50 | 45.93 | 15.82 | 58.59 | 52.13 | 57.16 |
| | V-RFT + PA$_P$ | **73.07** | **82.69** | **73.43** | **60.25** | **52.66** | **19.92** | **70.70** | **56.25** | **60.93** |
| | Δ | ↑ 0.00 | ↑ 0.80 | ↑ 1.56 | ↑ 0.75 | ↑ 6.73 | ↑ 4.10 | ↑ 12.11 | ↑ 4.12 | ↑ 3.77 |

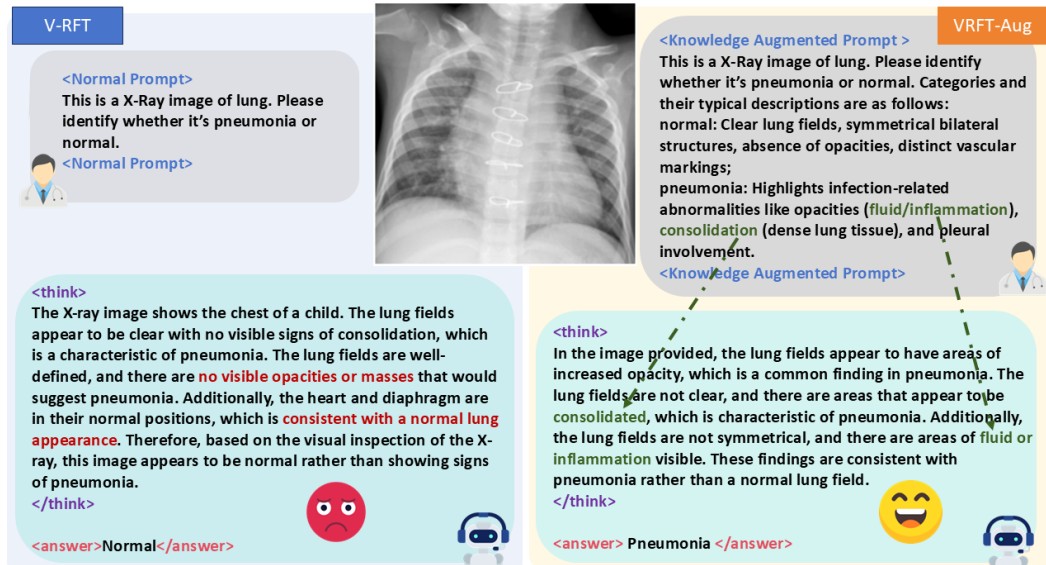

Figure 2: The effectiveness of our proposed perception augmentation on the prompt.

up to 2 epochs, with an initial learning rate of 1e-6 for both SFT and RL. The batch size is set to 1 per device, with 2-step gradient accumulation. Comprehensive details on the experimental settings and evaluation schemes are provided in Appendix B.2.

## 4.2 EXPERIMENTAL RESULTS

**Results on Contextual Augmentation.** For contextual augmentation, we compare V-SFT and V-RFT baselines on various few-shot settings with our V-RFT+PA$_P$ approach. As is shown in Table 1, both V-SFT and V-RFT can improve the model's performance under the few-shot settings, while our approach consistently outperforms all baselines and maintains a significant lead. With just 10 shots of data, our approach already delivers a boost by **+6.89%** compared with the V-RFT baseline. As the data amount increases, our approach achieves an average performance of **60.93%** in the 256-shot setting, **14.83%/3.77%** higher than V-SFT/V-RFT baselines. During the experiment, we have also noticed that contextual augmentation accelerates the training process. The phenomenon indicates that the model is incentivized to focus on feature-distinctive objects, thus enhancing its domain-specific perception and reducing the time required to learn task-relevant patterns.

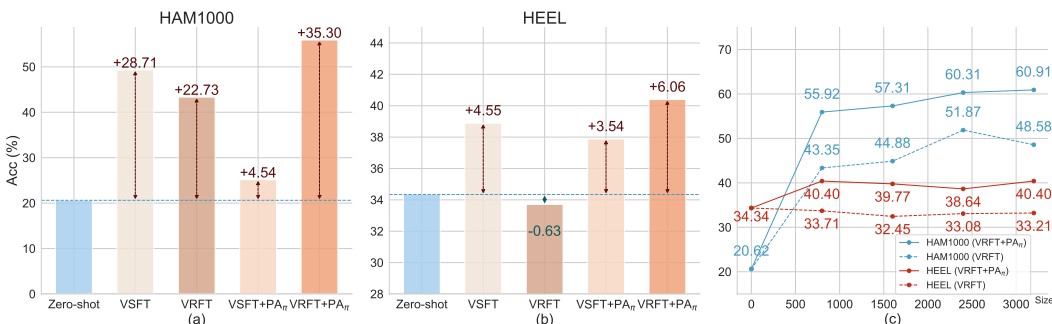

Figure 3: Performance comparison of different methods on the HAM10000 and HEEL. (a) and (b) show that VRFT + PA$\pi$ achieves the highest accuracy, with a +35.30% improvement on HAM10000. (c) demonstrates that performance of VRFT + PA$\pi$ improves with increasing training samples, reflecting enhanced perception capabilities. VSFT + PA$\pi$ and VRFT + PA$\pi$ are trained on bounding box prediction tasks (using SFT and GRPO, respectively) and evaluated on classification in a zero-shot manner, while V-SFT and V-RFT are directly trained for classification without localization.

Table 2: Comparison of the best results are highlighted in bold, while the second-best results are underlined. Note that with the exception of RetinaMNIST adopting MFRS reward, all other datasets utilize accuracy reward.

| Method | Breast | Pneumonia | OCT | Retina* | Derma | Tissue | Blood | OrganA | Average |
|---|---|---|---|---|---|---|---|---|---|
| Qwen2.5-VL-3B | 26.92 | 52.88 | 25.39 | 14.75 | 19.60 | 8.40 | 12.30 | 9.80 | 21.25 |
| V-SFT+ PA$_P$ | 58.33 | 76.12 | 55.46 | 52.75 | 49.57 | 17.96 | 70.50 | 42.18 | 52.86 |
| V-RFT + PA$_P$ | **73.07** | 82.69 | 71.87 | 60.25 | 52.66 | **19.92** | 70.70 | **56.25** | 60.93 |
| V-RFT + PA$_P$ + $\delta^+ R_{\text{recite}}$ | **73.07** | **83.49** | 66.79 | 49.00 | **56.02** | 12.50 | 70.31 | 51.70 | 57.86 |
| V-RFT + PA$_P$ + $\delta^- R_{\text{recite}}$ | **73.07** | 83.01 | **75.78** | **63.50** | 51.54 | 17.96 | **81.25** | 53.40 | **62.44** |

**Results on Implicit Knowledge Injection.** We evaluate the classification performance of five methods—zero-shot, V-SFT, V-RFT, V-SFT+PA$_\pi$ and V-RFT+PA$_\pi$ —on the HAM10000 and HEEL test sets. The zero-shot method refers to the Qwen2.5-VL-3B-Instruct model performing disease classification without any fine-tuning. As shown in Fig. 3 (a) and (b), it achieves 20.62% accuracy on HAM10000 and 34.34% on HEEL, indicating limited diagnostic performance and underscoring the need for downstream fine-tuning.

We then apply SFT and vanilla GRPO-based RFT, denoted as V-SFT and V-RFT, respectively. Both methods outperform zero-shot, validating the benefit of fine-tuning. Notably, V-RFT improves accuracy on HAM10000 by **+22.7%**, but slightly underperformed on HEEL (-0.63%). We find that the HEEL dataset suffers from data imbalance, and the less frequent classes have relatively more complex image features. We suspect that under complex or imbalanced data distributions, and in the absence of advanced techniques, the RFT may converge to suboptimal local patterns, overfitting to high-frequency and low-complexity features. In such cases, the simpler SFT may offer greater stability despite lacking reasoning capabilities. Next, we introduce V-SFT+PA$_\pi$ and V-RFT+PA$_\pi$, which incorporate a perception augmentation strategy via task-relevant training to inject implicit spatial knowledge. The model is first trained on localization tasks via SFT or RFT, followed by a zero-shot disease classification on the corresponding test sets. Notably, V-RFT+PA$_\pi$ demonstrates the most significant performance improvement across both datasets, with an impressive increase of **+35.30%** on the HAM10000 dataset. In contrast, although V-SFT+PA$_\pi$ also shows an improvement, the enhancement is less pronounced. These results indicate that training on the localization task to enhance the model's spatial perception ability is more effective in improving medical image classification performance. Moreover, it highlights that reinforcement learning, integrated during the inference process, further strengthens the model's anatomical localization perception. As observed in Fig. 3 (c), compared to V-RFT, the perception capability of V-RFT+PA$_\pi$ progressively improves as the model encounters more training samples, leading to continuous performance enhancement. In conclusion, we can assert that enhancing the model's anatomical localization perception capability significantly stimulates stronger performance in medical image classification.

Table 3: Performance variation between MFRS Reward and accuracy reward.

| Method | Retina | COVID-19 | Average |
|---|---|---|---|
| Qwen2.5-VL-3B | 14.75 | 17.64 | 16.20 |
| Qwen2.5-VL-7B | 21.00 | 20.26 | 20.63 |
| V-SFT | 50.00 | 19.60 | 34.80 |
| V-RFT | 59.50 | 20.26 | 39.88 |
| V-RFT+PA$_P$+R$_{acc}$ | 43.50 | 24.18 | 33.84 |
| V-RFT+PA$_P$+R$_{MFRS}$ | **60.25** | **30.06** | **45.16** |

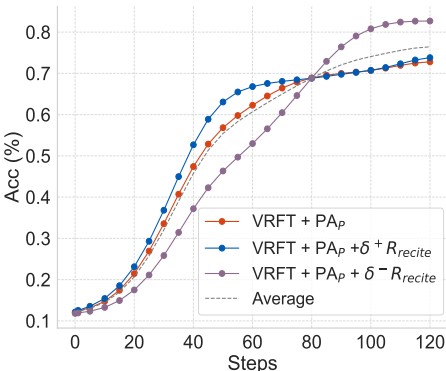

Figure 4: Performance variation on BloodM-NIST of different Recitation Reward settings.

**Results on Recitation Reward.** In this section, we compare our proposed V-RFT+PA$_P$ approach with different Recitation Reward modifications. In addition to quantitative results in Table 2, we also provide a curve graph of performance variation on BloodMNIST in Fig. 4. It can be observed from the figure that although repeated recitation of medical concepts can accelerate convergence to a sub-optimal plateau, it fails to achieve optimal performance in the long term. For other datasets in Table 2, the addition of positive Recitation Reward results in an average performance of 57.86%, 3.07% lower than the original proposed approach. The phenomenon indicates that over-reinforcing certain patterns may limit the model's flexibility and generalization. By contrast, a negative Recitation Reward can reduce the model's dependence on specific patterns. As is shown in Table 2, the average accuracy of negative $R_{recite}$ setting is 62.44%, creating a +1.51% improvement. Compared to the positive $R_{recite}$ setting, although the negative $R_{recite}$ setting causes a slight decline in DermaMNIST, TissueMNIST, and OrganAMNIST, the overall impact is only -0.74%, much smaller than the -3.59% decline observed with the positive $R_{recite}$ setting, highlighting the advantage of the negative setting in terms of model flexibility and generalization.

**Results on MFRS Reward.** To evaluate the validity of the MFRS Reward, we compare the classification performance of V-RFT+PA$_p$+R$_{MFRS}$ and V-RFT+PA$_p$+R$_{accuracy}$ in Table 3. It can be concluded that when replace $R_{MFRS}$ in Eq. (10) with $R_{accuracy}$, the average performance shows a noticeable decline from 45.16% to 33.84%, which even lags behind V-SFT/V-RFT by 0.96%/6.04%. These experimental results indicate that Vanilla RFT methods tend to suffer from the sparse reward problem Rengarajan et al. (2022); Dawood et al. (2023) due to the slight difference between categories in medical grade classification tasks. By allowing a "fuzzy" reward mechanism, the model can learn partial patterns by making near-correct predictions in the early stage instead of being trapped in invalid strategies.

## 5 CONCLUSION

In this study, we first identify the key challenges faced when applying RL-based training paradigm in medical visual recognition tasks. We argue that existing V-RFT methods must be improved from both the **perception** and **reasoning** perspectives to effectively adapt large vision-language models to the medical domain. Through extensive experiments, we show that there remains substantial room for improvement when applying GRPO-based V-RFT to medical scenarios. To address this, we propose a two-pronged enhancement framework: **VRFT-Aug**. On the **perception** side, we design methods that inject domain knowledge into the model explicitly by manipulating the prompt, and implicitly through cross-task training that embeds the implicit knowledge into the policy model. On the **reasoning** side, we design specialized reward functions tailored to the unique inference requirements of medical image recognition, including recitation reward function and multi-grade fuzzy reward function. As the first work targeting complex medical visual recognition tasks using reinforcement learning, we hope our training paradigm can inspire the broader research community and help pave the way toward the development of future medical reasoning models. While this method is specifically designed for medical tasks, the concept of prompt-based knowledge injection can potentially be extended to other domains. For instance, similar strategies could be applied to tasks requiring nuanced reasoning across diverse visual modalities, such as in autonomous driving or industrial inspections.

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

## A    APPENDIX

## B    TECHNICAL APPENDICES AND SUPPLEMENTARY MATERIAL

### B.1    DETAILED INFORMATION OF DATASETS USED FOR RELATIVE TASK

**Medical Image Classification** We use eight datasets from MedMNIST Yang et al. (2023), representing various imaging modalities, including X-Ray, Ultrasound, and CT: BreastMNIST, PneumoniaMNIST, OCTMNIST, RetinaMNIST, DermaMNIST, TissueMNIST, BloodMNIST, and OrganAMNIST. Most of these datasets contain over 15,000 images. For training efficiency, we randomly sample up to 256 images per class, except for RetinaMNIST and BreastMNIST, which have fewer than 1,500 images. This setup is treated as a 256-shot setting, with 10-shot and 20-shot settings derived similarly using a consistent test set.

**Fine-Grained Regional Classification.** To simulate the clinical workflow of locating and identifying lesions, we use two datasets: HAM10000 Tschandl et al. (2018) and Heel Taher & Özacar (2024). HAM10000 contains 10,015 dermoscopic images of seven skin lesion types, providing region of interest (ROI) masks without bounding boxes. We derive bounding boxes from the ROI edges. The Heel dataset consists of 3,956 X-ray images of foot lesions, designed for heel bone disease localization and classification.

**Severity Grading.** In addition to RetinaMNIST from MedMNIST, we also utilized Danilov's preprocessed dataset Danilov et al. (2022), which consolidates four publicly available datasets for COVID-19 and pneumonia classification. These datasets include Actualmed COVID-19 Chest X-ray agchung (2020a), COVID-19 Radiography Tawsifur Rahman (2022), COVID Chest X-Ray Cohen et al. (2020), and Figure1 COVID Chest X-ray agchung (2020b). Danilov's preprocessing standardizes these datasets and provides human-labeled severity scores ranging from 0 to 6, making them suitable for severity grading tasks. We combined these preprocessed datasets for consistent usage in our experiments.

### B.2    COMPREHENSIVE DETAILS OF EXPERIMENTAL SETTINGS AND EVALUATION METRIC

**Perception Augmentation Policies.**    As mentioned earlier, we use two approaches to enhance the model's perceptual capabilities in the medical imaging domain. One is by explicitly injecting medical prior knowledge into the model through prompt engineering to directly perform medical diagnosis, while the other is by transferring inherent knowledge through training on other tasks to improve the model's medical diagnostic ability.

- **Prior Knowledge Augmentation**: We train the model on MedMNIST datasets Yang et al. (2023) using two distinct prompt settings. The first setting only provides $\{Class\ Names\}$, while the second setting includes explicit knowledge injection, which provides both $\{Class\ Names\}$ and corresponding $\{Visual\ Attributes\}$. In addition to original dataset settings, we also apply SFT and RL on limited data, adopting 10-shot and 20-shot settings to evaluate the fine-tuned model's generalization ability. Note that for the overall reward $R$ we formulate it as $R = 0.9 \times R_{\text{accuracy}} + 0.1 \times R_{\text{formate}}$ by default. While for the RetinaMNIST dataset we adopt the MFRS reward for better performance, that is, $\hat{R} = 0.9 \times R_{\text{MFRS}} + 0.1 \times R_{\text{format}}$.

- **Visual Perception Augmentation**: We first train the model employing the R1 framework on the training set of the HAM10000 and Heel datasets, respectively, learning to localize specific regions, lesions, or organs. For example, detecting the bounding boxes for skin lesions in HAM10000 images and localizing the heel bone region in Heel images. Subsequently, without any additional training, we directly apply the model to the corresponding test sets for medical disease diagnosis in a zero-shot manner.

**Reasoning Augmentation via Reward Design.**

- **Recitation Reward**: We conduct two experiments by varying the value of $\delta$ in equation 7. When $\delta = 0.2$, the model is rewarded for repeating explicit knowledge during the thinking process. Conversely, when $\delta = -2$, the model is penalized for recitation.

- **MFRS Reward**: We train the model on the RetinaMNIST dataset in MedMNIST and the COVID-19 DatasetDanilov et al. (2022) using two reward settings to evaluate MFRS reward's validity. The difference is whether to replace $R_{\text{MFRS}}$ in euqation 10 with $R_{\text{accuracy}}$.

**Comparative Evaluation & Metric.** For the **Explicit Knowledge Injection** experiment, we primarily compare the performance of SFT and RL fine-tuning on the test sets, as well as the few-shot experiments. For the **Implicit Knowledge Injection** experiment, we compare the performance of RL and SFT in two approaches: (1) direct classification training, and (2) localization followed by direct classification. We use a metric similar to VQA choice accuracy. Each test sample consists of a medical question and a medical image, and the model must choose a diagnosis from a predefined list of lesion types. A correct diagnosis is made only when the model's prediction matches the ground truth. Finally, we evaluate the model's diagnostic performance by calculating the overall accuracy on the test set.

### B.3 Broader Impact

This paper presents work aimed at extending reinforcement learning fine-tuning into the domain of medical imaging. Our goal is to enhance model transparency by enabling visible reasoning processes during medical image interpretation. While this direction may have important implications for clinical AI applications, we believe no specific societal concerns need to be highlighted at this stage.

### B.4 Basic Rewards for Reinforcement Fine-Tuning

**Format Reward.** Following previous work Guo et al. (2025); Huang et al. (2025a); Shen et al. (2025a), we introduced format rewards to evaluate whether the model's generated output adheres to the expected structured format. Specifically, the model is enforced to enclose its thinking process between the `<think>...</think>` tags, include a bounding box within `<answer>{...[x1, y1, x2, y2]...}</answer>` for the detection task, or place the predicted label into `\boxed{...}` for the classification task, receiving 1 or 0 reward value based on compliance.

**Vanilla Accuracy Reward.** Detection task requires the model to provide the bounding box for a specific region, lesion, or organ in the medical image. Denote $GT^{\text{det}}$ as the ground truth bounding box, $O^{\text{det}}$ as the model output content, and $f_{det}$ as the function to extract the bounding box located by the VLM from its output content. The accuracy reward for detection task is defined as follows:

$$R_{acc}^{det} = \begin{cases} 1.0 & \text{if } \text{IoU}(GT^{\text{det}}, f_{det}(O^{\text{det}})) > threshold, \\ 0.0 & \text{otherwise.} \end{cases} \tag{11}$$

where IoU is the intersection over union metric, and the $threshold$ is typically set to 0.5 as default.

The vanilla accuracy reward for classification tasks is the most commonly used exact match, i.e., the model receives a reward score of 1 if the final answer exactly matches the ground truth when both are converted to lowercase; otherwise, the score is 0.

### B.5 Prompt Template for Localization Task

> **Prompt Template**
>
> This is a $\{data\ modality\}$ image of $\{lession/organ\}$. Please identify the category of the $\{lession/organ\}$ based on the image. Categories and their typical descriptions are as follows: $\{Class\ Names : Visual\ Attributes\}$. You FIRST think about the reasoning process as an internal monologue and then provide the final answer. The reasoning process MUST BE enclosed within <think> </think> tags. The final answer MUST BE put in \boxed{...}.

We need to construct training data for the localization task, using the following prompt template:

> **Prompt Template for Detection Task**
>
> Analyze the image and provide the bounding box for the *{target object}*. Ensure the bounding box accurately covers it and does not include too much unrelated areas. Output the bounding box in the format [x1, y1, x2, y2]. Generate your thinking process on how you determined the box. First output the thinking process in <think> </think> tags and then output the final answer in <answer> </answer> tags. Output the final answer in JSON format.

## C  LIMITATION

Our work is still limited to medical classification tasks, and has yet to explore fine-grained tasks such as segmentation. In addition, our current approach to knowledge injection lacks certain clinically grounded experiential knowledge. We plan to further investigate these directions in future work.

### C.1  PRINCIPLE OF BLEU METRIC

BLEU calculates similarity by comparing the overlap of n-grams between the candidate text and the reference text, making it particularly suitable for quantifying the similarity between the model's inference outputs and the prior knowledge.

