# OpenReview forum: "Improving Medical Visual Reinforcement Fine-Tuning via Perception and Reasoning Augmentation"
_ICLR.cc/2026/Conference — ICLR 2026 Conference Withdrawn Submission_

### Official Review · Reviewer_TuU5 · 2025-11-01

**Soundness:** 2
**Presentation:** 2
**Contribution:** 2
**Rating:** 4
**Confidence:** 2

**Summary:**

The paper proposes VRFT-Aug, an RL-based post-training framework for medical vision-language models (VLMs). It augments (i) perception via (a) prompt/context augmentation with domain attributes and (b) implicit knowledge injection by first learning localization with RL and then transferring to classification; and (ii) reasoning via (c) a recitation reward that encourages/discourages repeating injected knowledge and (d) a multi-grade fuzzy reward (MFRS) for ordinal grading. Experiments on several MedMNIST tasks show consistent improvements over SFT and vanilla RFT, with ablations on prompt/context/

**Strengths:**

# Strengths

1. Clear decomposition of failure modes (perception and reasoning) and mapping to concrete training knobs (prompt/context, localization transfer, reward shaping). The four components are easy to reproduce conceptually.

2. MFRS alleviates sparse rewards and gives notable gains over binary accuracy rewards on grading datasets.

3. Ablation shows the effectiveness of penalizing recitation, which can generalize better than rewarding it (positive), and is a non-obvious but actionable insight for RL recipes in VLMs.

**Weaknesses:**

# Weaknesses

1. Evaluation mainly on small/classification datasets; limited open-ended medical reasoning.
Many reported wins are on MedMNIST-style classification and a few fine-grained sets; these are simpler than full radiology VQA or report-generation and do not stress long-form reasoning or clinical justification as strongly as prior medical RL papers. The paper’s strongest novelty claims (recitation reward design/sign; localization-to-classification transfer) would be more compelling on harder, free-form medical VQA benchmarks where MedVLM-R1/Med-R1 already set a high bar.

2.  Rewarding/penalizing BLEU (n-gram) overlap with injected knowledge may (a) favor superficial copying or (b) punish legitimate paraphrase; the paper itself observes positive recitation converges to a “sub-optimal plateau,” underscoring metric-gaming concerns. Stronger signals (factuality/ontology verification or vision-grounded rationales) would better target reasoning quality.

**Questions:**

# Questions

1.	Do your benchmarks mostly test perception rather than reasoning? Please quantify and justify the “reasoning” claim.
2.	Is the recitation-induced sub-optimal plateau a consequence of your task suite and BLEU reward, or does it persist on free-form VQA?

---

### Official Review · Reviewer_BwRw · 2025-11-01

**Soundness:** 3
**Presentation:** 3
**Contribution:** 2
**Rating:** 4
**Confidence:** 5

**Summary:**

To address the limited improvement in reasoning ability of multimodal large vision-language models (LVLMs) after reinforcement learning (RL) in the medical domain, this paper introduces two key enhancements:

1.A two-stage knowledge injection process to enrich the domain-specific medical knowledge of LVLMs.

2.The design of new reward functions to improve the model’s reasoning capabilities.

More specifically, the paper proposes targeted modifications to three components of the original GRPO framework for optimizing LVLMs:

1.The prompts in the original RL training data are expanded using GPT, incorporating more medical terminology and clinical details. The experiments demonstrate that even by simply replacing the prompt, the method significantly alleviates the optimization bottleneck of GRPO for medical LVLMs, effectively injecting medical knowledge.

2.Before the main reinforcement learning phase, the policy model is trained on an auxiliary task that involves predicting bounding boxes based on medical image features, in order to improve its grounding in visual information.

3.On top of the standard GRPO reward functions such as accuracy reward and format reward, two additional rewards are introduced: recitation reward, which evaluates the extent to which the model’s reasoning path appropriately references the given prompt, and MFRS reward, a more lenient reward designed to better handle the verification of integer-type medical labels.

**Strengths:**

This paper addresses the limitations of the GRPO method in enhancing the reasoning capabilities of medical multimodal large vision-language models by proposing three targeted improvements including prompt expansion, auxiliary visual tasks and novel reward designs which demonstrate clear practical value.

**Weaknesses:**

In a previous conference, the reviewers had already raised concerns regarding related issues. However, compared to the previous conference, this paper has not addressed these issues, and the overall content remains consistent with the submission to the earlier conference. Therefore, the reviewers’ acknowledgment of the paper’s strengths and their concerns about its weaknesses are consistent with what was expressed in the previous conference：

1.Why does simply expanding the prompt lead to such significant improvements over the original GRPO, as shown in Table 1? In reinforcement learning, optimization signals originate only from the reward derived from the final outcome. Why is prompt modification able to achieve knowledge injection under an RL setting? Could the authors provide a brief mathematical explanation of this phenomenon and result?

2.Can the approach of knowledge injection through prompt expansion during reinforcement learning be generalized to domains beyond medicine?

3.Could the authors provide more experimental details about the reinforcement learning process, such as the dynamics of reward changes over time?

4.Regarding the RL baseline experiments, when comparing the effectiveness of the proposed method the question arises whether the policy model in the baseline was also trained with the bounding box prediction task. Considering that regional classification tasks are conceptually similar to bounding box prediction this raises concerns about the fairness of the experimental comparison.

5.Concerning the evaluation methodology, the paper states that many errors in the medical domain arise from knowledge gaps and methods like GRPO are generally used to enhance complex reasoning. The study could consider including tasks that more directly test complex reasoning abilities such as multimodal diagnostic scenarios in clinical settings rather than focusing on more traditional tasks like image classification and regional classification that conventional models can already handle.

6.The recitation reward seems particularly prone to reward hacking, such as the model repeating prompt content whenever the delta is positive. Could the authors provide additional experimental evidence or analysis to make this aspect more convincing? If the authors are able to address the above questions, the reviewer will consider raising the score.

**Questions:**

1.There is a lack of mathematical explanation for knowledge injection through prompt modification.

2.Some experimental details are not provided (especially the RL process).

If the authors are able to address the above questions, the reviewer will raise the score.

---

### Official Review · Reviewer_Kfgv · 2025-11-03

**Soundness:** 3
**Presentation:** 3
**Contribution:** 3
**Rating:** 6
**Confidence:** 4

**Summary:**

The paper introduces VRFT-Aug, a visual reinforcement fine-tuning framework that augments both perception and reasoning for large vision–language models in the medical domain. It enhances the standard V-RFT objective by (1) augmenting prompts with task-relevant contextual knowledge, (2) injecting implicit perceptual priors into the policy, (3) shaping the reward through a recitation reasoning term, and (4) shaping a multi-grade fuzzy reward that mitigates sparse-reward issues.

**Strengths:**

The paper makes a creative and well-motivated extension of RFT from LLMs to medical–language models. By bridging RFT and medical-language reasoning, this work could stand as a practical foundation for safe and interpretable medical AI system. The empirical improvements are consistent and meaningful. Its originality lies not in inventing a new algorithmic family, but in articulating a new decomposition of the RFT pipeline into perception, policy, and reward components, each augmented with domain-specific priors. Also, the algorithm yields clinically aligned reasoning behavior, not merely better accuracy.

I like the methodology. It is rigorous and shows empirical strong with principled reward shaping. The paper is well-structured. Guiding from motivation to formal definitions enables readers understand the algorithm very clearly. In addition, Figure 1 visualizes the modular design of each component.

**Weaknesses:**

W1. Incremental algorithmic novelty.

- The four augmentations (prompt, policy, recitation, fuzzy reward) are conceptually coherent but individually modest extensions of known techniques. Prompt engineering, auxiliary localization, imitation control, and fuzzy reward shaping. The work’s strength is integration rather than theoretical innovation. Given this work aims for medical purpose, I can understand this concatenation of existing techniques tho.

W2. Scalability and generalization not demonstrated.

- The experiments use Qwen2.5-VL-3B, a moderate-scale model; the method’s computational overhead and transfer behavior on larger LVLMs (e.g., InternVL-20B or Gemini-Vision) remain unexplored. Similarly, all datasets are relatively small and well-curated; testing on noisier real-world hospital data would better support claims of robustness.

**Questions:**

-

---

### Official Review · Reviewer_5Kig · 2025-11-10

**Soundness:** 2
**Presentation:** 2
**Contribution:** 2
**Rating:** 2
**Confidence:** 3

**Summary:**

This paper investigates the application of Reinforcement Fine-Tuning (RFT) to Large Vision-Language Models (LVLMs) for medical image analysis, a process Visual Reinforcement Fine-Tuning (V-RFT). The authors argue that standard V-RFT methods fail in the medical domain because they require both robust visual perception (to see subtle cues) and structured reasoning (to apply clinical logic).

**Strengths:**

The paper tackles a timely and significant problem: adapting reinforcement fine-tuning (RFT) for large vision-language models to the medical domain.

**Weaknesses:**

1. Disjointed Framework Evaluation
2. Limited Novelty of Components

**Questions:**

1.Please clarify the exact mechanism for the $PA_{\pi}$ evaluation? How does a model trained only on localization to output bounding boxes perform zero-shot classification?

2.Why was the full VRFT-Aug framework, combining all compatible components (e.g., $PA_p + PA_{\pi} + \delta^-R_{recite}$), never evaluated? The disjointed experiments make it difficult to judge the synergistic value of the proposed methods.

3.Which definition of the $R_{MFRS}$ reward is correct for a 2-class difference: $1/10$ 27or $0.0625$28?

4.How does the “recitation” mechanism affect the linguistic diversity of outputs during reasoning?

5.Limited generalization evaluation—the experiments focus mainly on MedMNIST-like datasets; real-world clinical validation or higher-resolution benchmarks would strengthen claims.

6.How sensitive is performance to the weighting parameters (λ, α, γ, δ) in the composite reward function?

7.Are there any ethical or bias considerations in using GPT-4o for generating domain-specific medical descriptions?

---

### Note · Authors · 2025-12-01

I have read and agree with the venue's withdrawal policy on behalf of myself and my co-authors.